# From 2D to 3D In Vitro World: Sonodynamically-Induced Prooxidant Proapoptotic Effects of C_60_-Berberine Nanocomplex on Cancer Cells

**DOI:** 10.3390/cancers16183184

**Published:** 2024-09-18

**Authors:** Aleksandar Radivoievych, Sophia Schnepel, Svitlana Prylutska, Uwe Ritter, Oliver Zolk, Marcus Frohme, Anna Grebinyk

**Affiliations:** 1Division Molecular Biotechnology and Functional Genomics, Technical University of Applied Sciences Wildau, Hochschulring 1, 15745 Wildau, Germany; alra9717@th-wildau.de (A.R.); sosc1766@th-wildau.de (S.S.); anna.grebinyk@desy.de (A.G.); 2Faculty of Health Sciences, Joint Faculty of the Brandenburg University of Technology Cottbus–Senftenberg, the Brandenburg Medical School Theodor Fontane and the University of Potsdam, 14476 Potsdam, Germany; oliver.zolk@mhb-fontane.de; 3Department of Plants Physiology, Biochemistry and Bionergetics, National University of Life and Environmental Science of Ukraine, Heroyiv Oborony Str., 15, 03041 Kyiv, Ukraine; psvit_1977@ukr.net; 4Institute of Chemistry and Biotechnology, Technical University of Ilmenau, 98693 Ilmenau, Germany; uwe.ritter@tu-ilmenau.de; 5Institute of Clinical Pharmacology, Brandenburg Medical School, Immanuel Klinik Rüdersdorf, 15562 Rüdersdorf, Germany; 6Deutsches Elektronen-Synchrotron DESY, Platanenallee 6, 15738 Zeuthen, Germany

**Keywords:** ultrasound, C_60_ fullerene, Berberine, sonodynamic therapy, apoptosis

## Abstract

**Simple Summary:**

Recently, sonodynamic therapy (SDT) has emerged as a promising non-invasive approach for treating cancer by activating sensitizers with ultrasound (US). In this context, we investigated C_60_ fullerene (C_60_) as a nanocarrier for the promising drug Berberine (Ber)—both potential aromatic sonosensitizers. The preferential mitochondrial accumulation of C_60_ and the proapoptotic effects of Ber also make the C_60_-Berberine nanocomplex (C_60_-Ber) a good candidate for direct induction of the intrinsic apoptotic cell death under US action. The in vitro research on C_60_-Ber can provide insights into novel, non-invasive cancer treatments. These findings lead to the development of targeted therapies with reduced side effects, inspire interdisciplinary collaboration, and open new avenues for drug delivery and cancer therapy research.

**Abstract:**

Objectives: The primary objective of this research targeted the biochemical effects of SDT on human cervix carcinoma (HeLa) and mouse Lewis lung carcinoma (LLC) cells grown in 2D monolayer and 3D spheroid cell culture. Methods: HeLa and LLC monolayers and spheroids were treated with a 20 µM C_60_-Ber for 24 h, followed by irradiation with 1 MHz, 1 W/cm^2^ US. To evaluate the efficacy of the proposed treatment on cancer cells, assessments of cell viability, caspase 3/7 activity, ATP levels, and ROS levels were conducted. Results: Our results revealed that US irradiation alone had negligible effects on LLC and HeLa cancer cells. However, both monolayers and spheroids irradiated with US in the presence of the C_60_-Ber exhibited a significant decrease in viability (32% and 37%) and ATP levels (42% and 64%), along with a notable increase in ROS levels (398% and 396%) and caspase 3/7 activity (437% and 246%), for HeLa monolayers and spheroids, respectively. Similar tendencies were observed with LLC cells. In addition, the anticancer effects of C_60_-Ber surpassed those of C_60_, Ber, or their mixture (C_60_ + Ber) in both cell lines. Conclusions: The detected intensified ROS generation and ATP level drop point to mitochondria dysfunction, while increased caspase 3/7 activity points on the apoptotic pathway induction. The combination of 1 W/cm^2^ US with C_60_-Ber showcased a promising platform for synergistic sonodynamic chemotherapy for cancer treatment.

## 1. Introduction

Due to random mutations in human cells, each patient case of cancer is unique [1,2,3]. The number of cancer patients is expected to grow by 12.8% in 2025 as compared to 2020 [4], which should drive the development of new efficient and accessible methods of cancer treatment. Sonodynamic therapy (SDT) is one of the promising approaches [5] that originated as a branch of photodynamic therapy (PDT). Instead of light used in PDT, ultrasound (US) induces the cytotoxic activity of chemical compounds in PDT [5,6]. These compounds, which are activated when exposed to ultrasound, are called sonosensitizers [5]. The sonoluminescence that occurs during US propagation in liquids after collapse of cavitating gas bubbles [7,8] excites sonosensitizer to its high energy state [9]. The excited sonosensitizer can either transfer an electron to oxygen, resulting in the formation of a superoxide anion radical O^−^_2_· or directly transfer energy to O^2^ to generate singlet oxygen ^1^O_2_ [9,10,11]. The superoxide anion radical then initiates free radical chain reactions, producing toxic hydroxyl radicals, hydrogen peroxide, and peroxynitrite [12]. These molecular oxygen-producing free radicals, also defined as reactive oxygen species (ROS), over certain levels can induce cell death [13] (Figure 1).

The advantage of SDT over PDT lies in US ability to penetrate tissues up to 10 cm in depth [14], while light, depending on the wavelength, reaches only up to 3 cm [15]. This ability makes SDT applicable for the treatment of deep-seated tumors [16]. The sonosensitizer can be designed to accumulate in tumor tissue; US can also be applied in a specific focused manner that together minimizes damage to non-malignant tissues surrounding the tumor. Compared to some conventional treatments (such as chemotherapy, radiotherapy, and surgery) that have limitations on administration frequency, SDT might be used for repetitive treatments [17,18]. SDT can also be combined with other therapies, such as PDT, chemotherapy, or radiotherapy, to enhance treatment efficiency [5,16,19].

In order to improve the therapeutic response to SDT, more effective and stable sonosensitizers are being developed. Various organic compounds with sonosensitizing capabilities have been transitioned from photodynamic therapy (PDT) to sonodynamic therapy (SDT), such as porphyrins [20], aminolevulinic acid [21], and Berberine [22]. Berberine (Ber, C_20_H_19_NO_5_, 5,6-dihydro-dibenzo[a,g]quinolizinium derivative) is an alkaloid extracted from various medicinal plants, including Indian barberry (*Berberis aristata)* and Japanese goldthread *(Coptis japonica*) [23]. Ber exhibits not only anti-inflammatory and proapoptotic effects by inhibiting mitochondrial respiration and modulating cell proliferation-related PI3K-AKT-mTOR signaling pathways [24,25], but also demonstrates sonosensitizing activity [24]. Berberine’s anti-inflammatory effects include reducing the expression of pro-inflammatory mediators including TNF-α, IL-1β, IL-17, MCP-1, MMP-2, and MMP-9 [25]. Ber is a promising candidate for SDT as it combines multiple therapeutic benefits, including anticancer activity, sonosensitizing effect, anti-inflammatory properties, and signaling pathway modulation. However, low bioavailability limits its medical applications [26].

In order to increase stability, treatment efficacy, and selectivity of further sonosensitizing agents towards cancer cells, carbon nanoparticles can be used as a base for drug delivery. In addition, carbon nanoparticles such as C_60_ fullerene, carbon dots, nanoribbons, and nanotubes [27,28,29,30] were shown to have high chemical and physiological stability as well as treatment efficacy in SDT. Hence, complexes with metal and carbon nanoparticles such as C-doped TiO_2_ [31], ferrite/carbon nanocomposite [32], fullerene-black phosphorus nanosheets [33], and fullerene/PMPC (poly(2-methacryloyloxyethyl phosphorylcholine)) complexes [34] have been demonstrated to effectively trigger ROS-mediated, compact apoptotic death in cancer cells following US treatment.

Fullerene C_60_ (C_60_) is one of the commonly explored carbon nanoparticles in PDT. C_60_ consists of 60 carbon atoms arranged in a series of interconnected hexagonal and pentagonal rings, forming a highly stable and symmetric sp^2.3^ hybridization structure [35,36]. C_60_ has very low solubility in water that limits its use in biomedical applications [37]. To enhance the solubility of C_60_ in aqueous solutions, methods such as derivatization and colloidal solutions are employed [38,39,40]. Functionalization serves to improve water solubility and enhance the biocompatibility of C_60_ by reducing the size of aggregates but also hinders C_60_’s interaction with cellular lipid membranes and alters its cellular uptake [38,40,41]. While C_60_ tends to aggregate in aqueous solutions, stable colloidal solutions can be formed [38]. Studies utilizing small-angle neutron scattering and atomic force microscopy have confirmed the stability of C_60_ aqueous colloidal solutions for up to six months [40]. The spectrum of sonoluminescence [42], overlaps with the absorbance spectrum of C_60_, which implies that the sonoluminescence can excite C_60_ and potentially trigger its cytotoxic photosensitizing activity [9]. The preferential mitochondrial accumulation of C_60_ [43,44] can directly induct the intrinsic apoptotic cell death under US action. In addition, C_60_ can be used as a nanocarrier for anticancer drugs. A pronounced antitumor effect of immobilized C_60_ such drugs as Doxorubicin [45,46], Cisplatin [47,48], and Ber [49] has been previously shown. Moreover, the C_60_-Doxorubicin nanocomplex demonstrated synergistic efficiency in combining PDT with 405 nm LED and chemotherapy [50].

Previously, we showed the promising anticancer effects of both C_60_ and Ber separately during SDT [22,28]. C_60_ and Ber decreased the cell viability of LLC cells to 56 ± 2% and 65 ± 2%, respectively. Similar results were obtained after treatment of HeLa cells; C_60_ and Ber decreased cell viability to 52 ± 2% and 57 ± 2%, respectively [22]. Relying on the observed sonosensitizing activities of both C_60_ and Ber and effective drug delivery with them complexed, this study is focused on the investigation of the C_60_-Berberine nanocomplex (C_60_-Ber) as a sonosensitizer towards cancer cells of different origin. For that, horizontal 2D cell culture was used as a common tool in cell biology and biomedical research. Furthermore, 3D cell culture can mimic the three-dimensional tissue structure and physiological conditions employed for in vitro modeling. The cytotoxic efficacy of the C_60_-Ber towards human cervix adenocarcinoma cells (HeLa) and Lewis lung carcinoma cells (LLC) in monolayers (2D cell culture) and spheroids (3D cell culture) were compared.

## 2. Materials and Methods

### 2.1. Chemicals

The pristine C_60_ aqueous colloid solution was synthesized at the Ilmenau University of Technology using continuous ultrasound sonication for C_60_ transfer from toluene to water according to Ritter et al. [40] (Ilmenau, Germany). Dihydroethidium (DHE) and 3-(4,5-dimethylthiazol-2-yl)-2,5-diphenyl tetrazolium bromide (MTT) were obtained from Biomol GmbH (Hamburg, Germany), while Berberine was procured from Sigma-Aldrich Co. (St. Louis, MO, USA). The Mitochondrial ToxGlo™ Assay and Caspase-Glo^®^ 3/7 Assay kits were supplied by Promega GmbH (Walldorf, Germany). Dulbecco’s modified Eagle’s medium (DMEM), phosphate-buffered saline (PBS), fetal bovine serum (FBS), penicillin/streptomycin, l-glutamine, and trypsin were sourced from PAN-Biotech GmbH (Aidenbach, Germany).

### 2.2. C_60_-Berberine Complex Synthesis

The preparation of the C_60_-Ber nanocomplex was performed in an aqueous solution and confirmed by the UV–Vis spectroscopy with SQ-4802 (Unico, Waltham, MA, USA) according to the protocol [49]. Briefly, the C_60_ aqueous colloid solution and Ber were mixed in a 1:1 molar ratio (200:200 μM). The resulting mixture of C_60_ and Ber was treated in the ultrasonic disperser Sonorex RK 31 (Bandelin Electronic, Berlin, Germany) for 20 min and then stirred for 24 h at room temperature. The hydrodynamic diameter of particles, polydispersity index (PDI), and zeta potential values for the C_60_-Ber at room temperature are 114 ± 2 nm, 0.42 ± 0.02, and −20.6 ± 0.5 mV, respectively [49]. The structure of the nanocomplex represents a face-to-face orientation of the Ber and C_60_’s aromatic surfaces, with a distance between molecules of 0.328 nm [49].

### 2.3. Cell Culture and Treatment with Agents under Study

The human cervix adenocarcinoma cell line (HeLa) was generously supplied by the Division of Gastroenterology, Infectiology, and Rheumatology, Charité Universitätsmedizin (Berlin, Germany). The Lewis lung carcinoma cell line (LLC) was acquired from Tebu-Bio GmbH (Offenbach, Germany). Both HeLa and LLC cells were cultured in DMEM, supplemented with 10% FBS, 1% penicillin/streptomycin, and 2 mM l-glutamine.

Cells were cultured in monolayers in 25 cm^2^ flasks at 37 °C, with 5% CO_2_ in a humidified incubator binder (Tuttlingen, Germany). Treatment with a 0.5% trypsin solution was used to detach cells. The number of viable cells was counted upon 0.1% trypan blue staining with a Roche Cedex XS analyzer (Basel, Switzerland).

#### 2.3.1. Preparation of Cell Monolayers

For experiments, cells were seeded in Petri dishes Ø = 35 mm (Sarstedt AG & Co. KG, Nümbrecht, Germany) at 1.5 × 10^5^ cells per Petri dish and incubated for 24 h. Then, media was changed to 1% FBS DMEM containing 20 µM C_60_-Ber nanocomplex, a mix of 20 µM C_60_ and 20 µM Ber, 20 µM C_60_ or 20 µM Ber, and 24 h before US exposure. The control groups of cells were treated with an equal volume of sterile water. Prior to US exposure, the volume of cell culture media in Petri dishes was increased to 7.5 mL. After US exposure, cells were transported from Petri dishes to a transparent 96-well plate (Sarstedt AG & Co. KG, Nümbrecht, Germany) for MTT assay, a black 96-well plate (Thermo Fisher Scientific Inc., Berlin, Germany) for ROS measurement, and a white 96-well plate (Thermo Fisher Scientific Inc., Berlin, Germany) for ATP and caspase activity measurement in concentration 10^4^ cells in 100 µL of cell culture media per well.

#### 2.3.2. Preparation of Cell Spheroids

For experiments, cells were seeded in round bottom low-attachment 96-well plates (Thermo Fisher Scientific Inc., Berlin, Germany) at 10^4^ cells per well and incubated for 24 h. Then, media was changed to 1% FBS DMEM containing 20 µM C_60_-Ber nanocomplex, a mix of 20 µM C_60_ and 20 µM Ber, 20 µM C_60_ or 20 µM Ber, and 24 h before US exposure. The control groups of cells were treated with an equal volume of sterile water. Before US exposure, the volume of cell culture media in wells was increased to 400 µL. After US exposure, cell spheroids were transported to a black 96-well plate (Thermo Fisher Scientific Inc., Berlin, Germany) for ROS measurement or a white 96-well plate (Thermo Fisher Scientific Inc., Berlin, Germany) for ATP and caspase activity measurement.

### 2.4. Ultrasound Exposure

The set-up for the US exposure is shown schematically in Figure 2. The US therapy unit DIGI (Strive Enterprises, Haryana, India) was driven at 1 MHz with 1 W/cm^2^ in continuous wave mode for 30, 60, or 90 s. The distance between the transducer and well or Petri dish bottom was 10 mm.

### 2.5. Cell-Based Assays

#### 2.5.1. Cell Viability

Cell viability was determined with an MTT assay [51] at 24 h after US treatment. Briefly, cells (10^4^/well) were incubated for 2 h at 37 °C in the presence of 0.5 mg/mL MTT. The diformazan crystals in wells were dissolved with dimethylsulfoxid (DMSO), and absorbance was determined at 570 nm using a microplate reader, Tecan Infinite M200 Pro (Tecan Trading AG, Männedorf, Switzerland). Phase contrast microscopy was performed with the Keyence Microscope BZ-9000 BIOREVO (Keyence Corporation, Osaka, Japan) to observe the visual effects of treatment. Images were captured and processed with BZ-II Viewer (Keyence Corporation, Osaka, Japan).

#### 2.5.2. Intracellular Reactive Oxygen Species Generation

To assess ROS production, Dihydroethidium (DHE) was used. A 10 mM stock solution of DHE was prepared in DMSO, stored at −20 °C, and diluted with PBS just before use. Cells (10⁴/well) were washed once with PBS, then treated with 10 µM DHE solution and incubated for 30 min. Fluorescence (λ_ex_ = 495 nm, λ_em_ = 585 nm) was measured using the Tecan Infinite M200 Pro microplate reader (Tecan Trading AG, Männedorf, Switzerland).

#### 2.5.3. Intercellular ATP Content

Cell culture media in plates was changed to 50 µL glucose-free DMEM after US treatment. Cellular ATP levels were measured using the Mitochondrial ToxGlo™ Assay Kit (Promega GmbH, Walldorf, Germany) following the manufacturer’s instructions. The plates were brought to room temperature, and 50 µL of ATP detection reagent was added to each well. After shaking at 600 rpm for 1 min, the luminescence intensity was recorded using the Tecan Infinite M200 Pro microplate reader.

#### 2.5.4. Caspase 3/7 Activity

Caspase 3/7 activity was measured over a 3 h period after US exposure using the Caspase-Glo^®^ 3/7 Activity Assay Kit (Promega GmbH, Walldorf, Germany) following the manufacturer’s protocol. In summary, the plates were removed from the incubator and allowed to reach room temperature for 30 min. After treatment, an equal amount of Caspase-Glo 3/7 reagent was added, and the mixture was shaken at 300 rpm for 1 min. The plates were then incubated at room temperature for 30 min, and the luminescence of each sample was measured using the Tecan Infinite M200 Pro microplate reader.

### 2.6. Statistics

All experiments were carried out with a minimum of three replicates. Data analysis was performed with GraphPad Prism 7 (GraphPad Software Inc., San Diego, CA, USA) and STATISTICA 12 (StatSoft GmbH, Hamburg, Germany). A paired Student’s *t*-test was performed to test whether the difference between the responses of control and irradiated US cells is statistically significant or not. The significance level was set at *p* < 0.01.

The combination index (CI), calculated according to the Chou–Talalay method [52] with CompuSyn (ComboSyn Incorporated, Paramus, NJ, USA), was used to evaluate sonodynamic interactions between sonoexcited C_60_ and Ber in cells treated with C_60_-Ber and irradiated with 1 MHz 1 W/cm^2^ US during 30, 60, and 90 s. The following equation was used:$$CI=\frac{{\left(D\right)}_{1}}{{\left(D_{x}\right)}_{1}}+\frac{{\left(D\right)}_{2}}{{\left(D_{x}\right)}_{2}}$$
where (*D_x_*)_1_ is the concentration of free sonoexcited Ber that inhibited cell viability to x%; (*D_x_*)_2_ is the concentration of free sonoexcited C_60_ that inhibited cell viability to x%; (*D*)_1_ and (*D*)_2_ are the concentrations of Ber and C_60_ in the C_60_-Ber nanocomplex that inhibited cell viability to x% after sonoexcitation. The CI values < 1, =1, and >1 indicated synergistic, additive, or antagonistic interaction, respectively.

## 3. Results and Discussion

Cell culture is an indispensable tool in cancer research, allowing the maintenance and proliferation of cells outside of the human body. It provides a controlled environment to study cancer cell responses to the proposed experimental treatment. Previously published studies showed that C_60_ can be predominantly accumulated in mitochondria and exhibit pronounced cytotoxic effects under US exposure [28,44,53]. Previously, we demonstrated sonosensitizing effects of C_60_ and Ber separately towards LLC and HeLa monolayers [22]. The treatment of LLC cells with C_60_-Ber was followed by more intensive intracellular Ber accumulation as compared with free Ber [49]. C_60_-Ber decreased the viability of LLC cells to 76 ± 4%, while free Ber only to 46 ± 6% after 24 h of incubation [54]. In addition, the efficacy of C_60_–Ber was shown in vivo. The indexes of both tumor mass and tumor weight in LLC tumor-bearing C57Bl male mice treated with C_60_-Ber decreased to 50%, while separately C_60_ and Ber demonstrated no significant effect [54]. Taken together, C_60_ and Ber sonosensitizing activities and improved delivery of Ber in its nanocomplex with C_60_, C_60_-Ber demonstrated promising efficacy under US exposure towards cancer SDT. Cell viability, ROS, ATP, and caspase 3/7 activation levels of C_60_ or Ber separately towards spheroids of LLC and HeLa cells are demonstrated in Appendix A (Figure A1).

Our findings were based on the viability of cancer cells following US treatment (Figure 3). To evaluate whether the sonodynamic treatment of HeLa and LLC cells incubated with C_60_-Ber can enhance toxic effects, cell viability was initially assessed using the MTT assay in 2D cell monolayers. Following a 24 h incubation with 20 μM C_60_-Ber, cells were exposed to 1 MHz US. The MTT test was used to estimate the cell viability of treated LLC and HeLa cells 24 h after US exposure. Sonication alone at low (30 s) and middle (60 s) doses had no significant viability changes in both LLC and HeLa cells. The longest 1 MHz US sonication within 90 s, applied in the experiments, decreased cell viability of LLC and HeLa cells to 68 ± 12% and 72 ± 7%, respectively. Such cell viability drop could be a result of direct cell membrane damage by US-induced streaming of surrounding liquid, cavitation, and oscillation of the well bottom [55,56,57]. Therefore, ≥90 s US treatment durations were omitted in the following experiments. C_60_-Ber alone did not demonstrate significant cytotoxicity towards cancer cells. However, after irradiation with 60 s 1 MHz 1 W/cm^2^ US, C_60_-Ber decreased cell viability of LLC and HeLa cells to 37 ± 4% and 32 ± 8%, respectively. Sonicated C_60_ and Ber mixture decreased cell viability of LLC and HeLa cells to 56 ± 2% and 56 ± 7%, respectively, which was equal to the sonodynamic cytotoxicity of free C_60_ previously shown in [22]. Figure 4 shows phase contrast microscopy images of cell monolayers after treatment. Decreased cell density and apoptotic body formation were observed after treatment with C_60_-Ber and 60 s of 1 MHz 1 W/cm^2^ US, whereas US without sonosensitizer or each sonosensitizer alone did not visually affect cells.

Cell spheroids mimic the three-dimensional architecture of tissues more accurately than cell monolayers and offer a better representation of the tumor tissue microenvironment, including conditions such as hypoxia and nutrient gradients, potentially reducing the gap between in vitro and in vivo biomedical models.

When comparing 3D cell spheroids with 2D monolayers, a similar trend in cell viability decrease was found after sonodynamic treatment (Figure 4). Sonication with 1 MHz 1 W/cm^2^ US during 30 and 60 s demonstrated no harmful effect on cell viability, while 90 s of sonication decreased cell viability to 83 ± 4% and 85 ± 4% of LLC and HeLa spheroids, respectively. A higher resistance of cell spheroids to US in comparison to cell monolayers can be linked to the lower effect of the well’s bottom oscillation in 3D cell culture conditions. The C_60_-Ber nanocomplex, a mixture of C_60_ and Ber, C_60_ or Ber separately did not demonstrate significant cytotoxicity towards both LLC and HeLa cancer cell spheroids. Although, after sonication with 1 MHz 60 s 1 W/cm^2^ US, C_60_-Ber decreased cell viability to 37 ± 5% and 38 ± 4% of LLC and HeLa cells in spheroids, respectively. C_60_ and a mixture of C_60_ and Ber demonstrated significantly non-different cytotoxicity levels towards cell spheroids and cell monolayers. Meanwhile, LLC and HeLa cells in spheroids showed higher resistance to Ber than their monolayers. The viability of LLC and HeLa spheroids treated with Ber and US dropped to 77 ± 9% and 72 ± 6%, respectively. Figure 4 presents phase-contrast microscopy images of cell spheroids after treatment with sonosensitizers and US. 1 MHz 1 W/cm^2^ US for 60 s did not affect cell spheroids; however, a combination of C_60_-Ber and US caused minor deformation of spheroid shape and cell disintegration into the media.

The combination index (CI) was calculated to assess whether Ber nanocomplexation with C_60_ had any synergistic effect after sonoexcitation (Table 1). Calculation of CI revealed the synergistic effect of the treatment with sonoexcited complexed C_60_ and Ber towards both cancer monolayers and spheroids. CI indicated strong synergism towards LLC cell monolayers after sonoexcitation with 1 MHz 1 W/cm^2^ US during 30, 60, and 90 s. However, the effectiveness of synergy dropped towards LLC cell spheroids; meanwhile, towards HeLa cell monolayers and spheroids, it was equal.

Excessive intracellular ROS production by sonosensitizing molecules can disrupt the balance between ROS generation and scavenging. This imbalance leads to oxidative cellular damage and activation of redox-sensitive signaling pathways involved in apoptosis. Hence, the production of intracellular ROS is essential for achieving the cytotoxic effect of the sonodynamic treatment [28,44,58]. The evaluation of ROS generation was performed with the ROS-sensitive fluorescence dye DHE. DHE can be oxidized by both superoxides to form 2-hydroxyethidium (2-OH-E+) and non-specific oxidation by hydrogen peroxide and hydroxyl radicals to form ethidium (E+) [59]. US exposure without the C_60_-Ber nanocomplex or a mixture of C_60_ and Ber had no significant effect on intracellular ROS levels. Meanwhile, sonication together with C_60_-Ber increased ROS levels to 213 ± 3% and 398 ± 73% in monolayer LLC and HeLa cells, respectively (Figure 5). Irradiation of LLC and HeLa spheroids induced similar ROS production as in monolayers that were equal to 208 ± 16% and 430 ± 34%, respectively. Intracellular levels of ROS were significantly higher in HeLa cells in comparison with LLC cells after treatment. Our previous study revealed a similar tendency in ROS levels between LLC and HeLa cells treated with 1 MHz 1 W/cm^2^ US with C_60_ or Ber [28]. These results can be related to increased activity of the antioxidant system in lung cancer cell lines [60], thus excessive ROS production was hindered more efficiently in LLC cells as compared to HeLa.

Excessive ROS production can directly impair ATP production by damaging mitochondrial respiratory complexes, further exacerbating mitochondrial dysfunction [61]. ATP levels decrease can contribute to apoptosis progress, inducing cellular energy depletion and metabolic collapse [62,63]. On the other hand, since apoptosis relies on ATP, a complete ATP level drop may result in non-programmed necrotic cell death and trigger an inflammatory response [64,65]. The ATP level was evaluated to characterize mitochondrial activity. Neither US nor C_60_-Ber separately had an effect on the ATP level of cancer cell monolayers or spheroids (Figure 6). However, after treatment with C_60_-Ber and 1 MHz 1 W/cm^2^ US for 60 s, ATP levels in LLC and HeLa cell monolayers decreased to 25 ± 1% and 42 ± 6%, respectively, which indicated its high toxicity towards mitochondria. Spheroids of LLC and HeLa cells demonstrated higher ATP levels than those grown in monolayers. The ATP levels of LLC and HeLa spheroids were equal to 67 ± 6% and 66 ± 4%, respectively, after treatment with C_60_-Ber and 1 MHz 1 W/cm^2^ US during 60 s. ATP level decrease can be related to mitochondrial fragmentation during apoptosis [63]. As maintenance of ATP level is much preferable for apoptotic cell death [64,65], the obtained results can indicate that cells, grown both in monolayers and spheroids, were prone to undergo apoptotic cell death.

The intrinsic pathway of apoptosis is a tightly regulated process where ROS generation, ATP depletion, and caspase 3/7 activation are interconnected events. ROS-mediated oxidative stress does not only induce ATP depletion but also can activate proteases of the caspase family [64,65]. The activation of caspase 3/7 is a key event in the apoptotic process [64]. The possible proapoptotic effect of the combinative treatment of C_60_-Ber with 1 MHz 1 W/cm^2^ US towards cancer cell monolayers and spheroids was evaluated with caspase 3/7 activity. Without sonoexcitation, C_60_-Ber increased caspase 3/7 activity in LLC and HeLa monolayers to 225 ± 25% and 253 ± 3%, respectively. Meanwhile, sonoexcited C_60_-Ber nanocomplex increased activity of caspase 3/7 by 404 ± 45% and 438 ± 51% in the cell monolayer of LLC and HeLa cells, respectively (Figure 7). The obtained data showed lower caspase 3/7 induction in LLC and HeLa spheroids after treatment with C_60_-Ber and US in comparison to the respective cell monolayers, with activity equal to 189 ± 17% and 247 ± 25% correspondingly. The treatment only with C_60_-Ber demonstrated a significantly lower increase in caspase 3/7 activity in cancer cell monolayers and spheroids in comparison with treatment with both US and C_60_-Ber. Meanwhile, the data revealed higher caspase 3/7 activity in cancer cell monolayers than in spheroids of both LLC and HeLa cells on 215% and 191%, respectively.

The obtained results showed that cancer cell spheroids were more resistant to the treatment effects than the cell monolayers. This is consistent with Uematsu et al., who showed higher chemo-sensitivity to Daunorubicin, Docetaxel, and Arsenic Disulfide of monolayers in comparison with spheroids of human breast cancer MCF-7 cells [66]. The same tendency was observed by Breslin and O’Driscoll with HER2-positive breast cancer cell lines treated with Docetaxel and Neratinib [67]. Chen et al. demonstrated less sensitivity of breast cancer spheroids than monolayers during PDT with cisplatin and red LED light [68]. Increased resistance of cancer cell spheroids to treatment can be connected with the hypoxic conditions in the core of spheroids and the inducement of hypoxia-inducible factor 1 (HIF-1) and CAIX that are linked to drug resistance [69]. Overall, spheroids may provide more accurate predictions of drug efficacy, toxicity, and pharmacokinetics compared to traditional 2D models. Therefore, the utilization of spheroids can be classified as a superior model for treatment screening, which replicates in vivo environments better than conventional monolayers. In vivo models are generally more resistant to cancer treatment compared to in vitro models due to several factors that more accurately mimic physiological conditions, as interactions with the extracellular matrix and surrounding stromal cells provide survival signals and create a protective environment against treatments [70].

The aqueous solution of C_60_ and its nanocomplexes have attracted attention in the field of cancer therapy due to C_60_’s potential as a sonosensitizer for SDT [22,28,33,34]. Nguyen et al. reported HeLa cell viability dropped to 10% under treatment with C_60_/PMPC (poly(2-methacryloyloxyethyl phosphorylcholine)) complexes and sonication for 3 min at 100 W and 42 ± 6 kHz [34]. C_60_ with covalently functionalized few-layer black phosphorus nanosheets under 1 MHz 1 W/cm^2^ US demonstrated treatment-induced cytotoxicity towards mouse 4T1 breast cancer cells, whose survival rate was decreased to ≈35% [33]. The presented study reveals the cytotoxic effects of the proposed sonodynamic treatment of 1 MHz US in combination with the C_60_-Ber towards cancer cells of lung and cervix tissue origin both in 2D and 3D cell culture conditions. The observed cytotoxicity was manifested through oxidative stress, ATP level drop, and caspase 3/7 induction. Meanwhile, prolonged excessive ROS generation by a high dose of applied US or sonosensitizer can lead to necrosis [71,72]. Nejda et al. reported that treatment of oral squamous cell carcinoma with 1 MHz 73 W/cm^2^ US and TiO_2_ nanoparticles resulted in necrosis, while lower doses of US led to apoptosis [73]. Tarkovsky et al. showed considerable necrotic tissue damage after treatment with modified chlorin e6 and 1 MHz 0.7 W/cm^2^ US [74]. The apoptotic or necrotic response correlates with the US intensity and concentration of the drug, indicating a US-drug dose-dependent relationship. Apoptosis is more preferable in cancer treatment strategies than necrosis due to its selective nature, minimal tissue damage, immunological properties, association with treatment response, and ability to maintain tissue integrity and homeostasis [64,65]. Hence, targeting apoptotic pathways has become a central focus of therapeutic approaches against cancer, including SDT. Given the promising prooxidant proapoptotic effects of synergistic treatment of cells both in 2D and 3D with sonoexcited C_60_-Ber nanoxomplexes, their pharmacodynamics and pharmacokinetics must be thoroughly characterized through both in vitro and in vivo studies to evaluate potential human health risks and benefits.

The observed cellular responses to the C_60_ nanocomplex-based SDT exhibit a similar pattern of PDT in vitro effects, where light is being used for C_60_ photosensitizing activity induction. Thus, after irradiation of human leukemic CCRF-CEM cell monolayers with 405 nm LED at 10 J/cm^2^ in the presence of C_60_-Doxrubicin nanocomplex, cell viability was decreased to ≈20%, while dark control of C_60_-Doxorubicin nanocomplex treatment was less efficient with ≈40% of detected cell viability [50]. SDT and PDT are both non-invasive cancer treatment modalities that utilize US or light in combination with sensitizing molecules to induce local cytotoxic effects. Except for deeper tissue penetration, SDT offers several unique benefits compared to PDT. Light used in PDT can be scattered or absorbed by surrounding tissues, limiting its ability to effectively penetrate deep tumor tissues in comparison with US [75]. Light can also cause skin photosensitivity and adverse reactions to sunlight in treated patients. SDT, on the other hand, does not pose the same risk of skin photosensitivity since it utilizes US [17]. These advantages make SDT a promising approach for the treatment of a wide range of cancers, particularly those that are difficult to treat using conventional therapies.

## 4. Conclusions

The combination of 1 W/cm^2^ US with C_60_-Ber nanocomplex showcased a promising platform for synergistic sonodynamic chemotherapy for cancer treatment. The demonstrated intensified ROS generation, ATP level drop, and caspase 3/7 activity induction resulted in a viability decrease in LLC and HeLa cells, cultured both in 2D and 3D in vitro conditions and treated with sonicated C_60_-Ber nanocomplex. These findings evidenced the prooxidant proapoptotic activity of the proposed cancer cell treatment strategy.

## Figures and Tables

**Figure 1 cancers-16-03184-f001:**
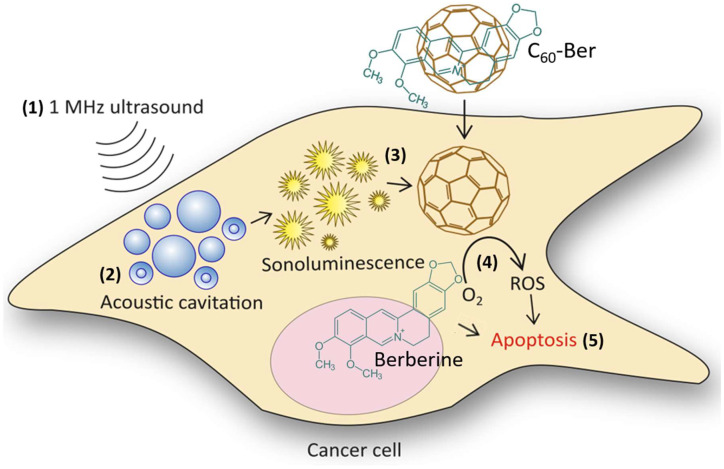
Combination of sonodynamic therapy and chemotherapy with a C_60_-Ber nanocomplex: cell exposure to high-frequency US (1); acoustic cavitation (2); C_60_ excitation with sonoluminescence (3); ROS generation by excited C_60_ (4); apoptotic death caused by generated ROS and co-delivered Ber (5).

**Figure 2 cancers-16-03184-f002:**
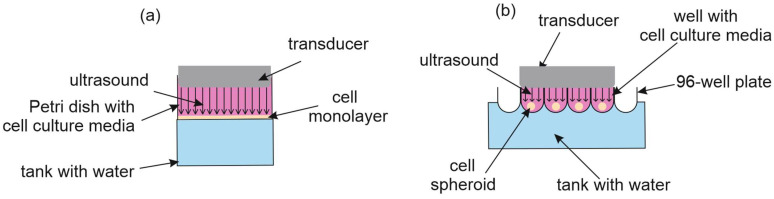
Diagram of the ultrasound exposure set-up for cells cultured in 2D (**a**) and 3D (**b**) conditions.

**Figure 3 cancers-16-03184-f003:**
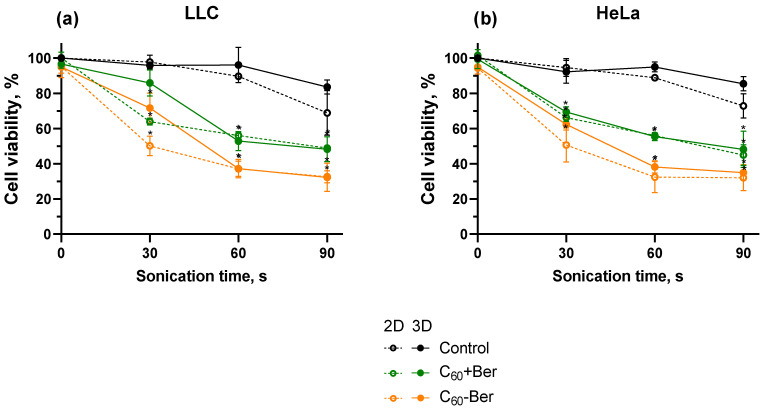
Viability of LLC (**a**) and HeLa (**b**) cell monolayers (2D) and spheroids (3D) incubated in the presence of 20 µM C_60_-Ber nanocomplex (C_60_-Ber) or mixture of 20 µM C_60_ and 20 µM Ber (C_60_ + Ber) and treated with 1 MHz ultrasound (US); * *p* ≤ 0.01 in comparison with the viability of cells treated with the respective duration of US.

**Figure 4 cancers-16-03184-f004:**
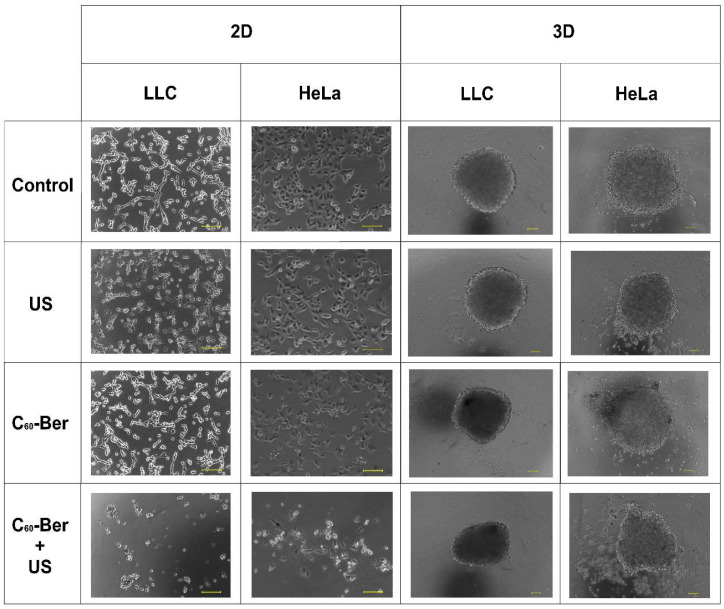
Phase contrast microscopy images of monolayers (2D) and spheroids (3D) of LLC and HeLa cells incubated in the presence of 20 µM C_60_-Ber nanocomplex and irradiated with 60 s 1 MHz 1 W/cm^2^ ultrasound (US) in the “transducer in well” set-up; scale bar is 100 µm.

**Figure 5 cancers-16-03184-f005:**
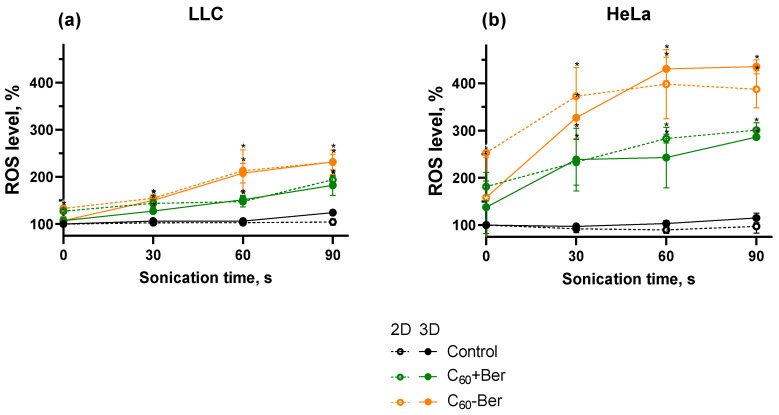
ROS level of LLC (**a**) and HeLa (**b**) cell monolayers (2D) and spheroids (3D) incubated in the presence of 20 µM C_60_-Ber nanocomplex (C_60_-Ber) or mixture of 20 µM C_60_ and 20 µM Ber (C_60_ + Ber) and treated with 1 MHz ultrasound (US); * *p* ≤ 0.01 in comparison with the ROS level of cells treated with the respective duration of US.

**Figure 6 cancers-16-03184-f006:**
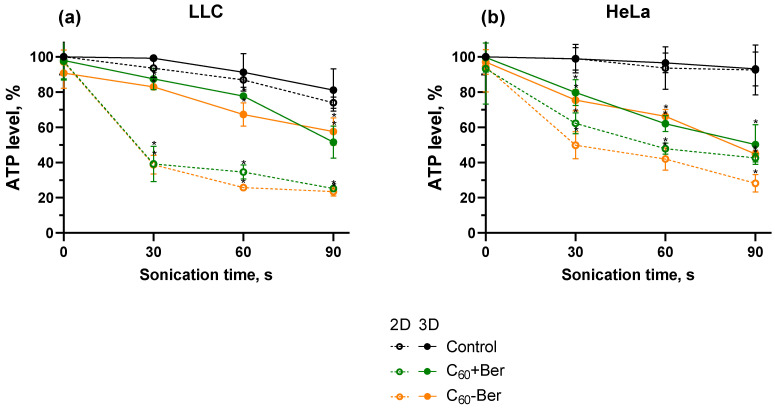
ATP level of LLC (**a**) and HeLa (**b**) cell monolayers (2D) and spheroids (3D) incubated in the presence of 20 µM C_60_-Ber nanocomplex (C_60_-Ber) or mixture of 20 µM C_60_ and 20 µM Ber (C_60_ + Ber) and treated with 1 MHz ultrasound (US); * *p* ≤ 0.01 in comparison with the ATP level of cells treated with the respective duration of US.

**Figure 7 cancers-16-03184-f007:**
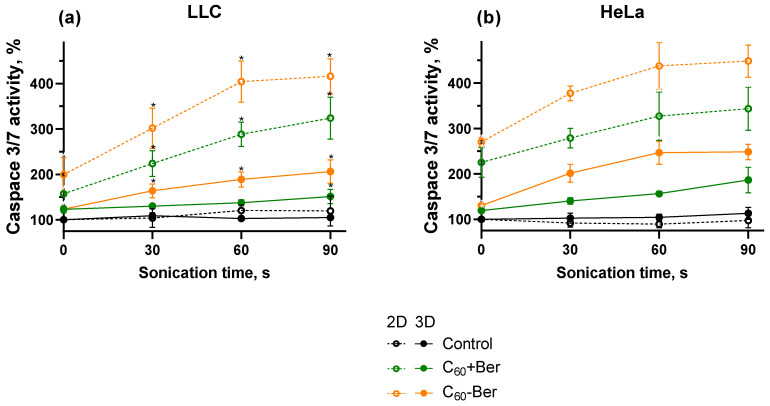
Caspase 3/7 activity of LLC (**a**) and HeLa (**b**) cell monolayers (2D) and spheroids (3D) incubated in the presence of 20 µM C_60_-Ber nanocomplex (C_60_-Ber) or mixture of 20 µM C_60_ and 20 µM Ber (C_60_ + Ber) and treated with 1 MHz ultrasound (US); * *p* ≤ 0.01 in comparison with the caspase 3/7 activity of cells treated with the respective duration of US.

**Table 1 cancers-16-03184-t001:** Combination index (CI) of interaction between the sonotoxic effects of C_60_ and Ber.

CI	LLC	HeLa
2D	0.25 (strong synergism)	0.39 (synergism)
3D	0.44 (synergism)	0.40 (synergism)

## Data Availability

The datasets used and analyzed during the current study are available from the corresponding author upon reasonable request.

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
