# Peer review of "From 2D to 3D In Vitro World: Sonodynamically-Induced Prooxidant Proapoptotic Effects of C60-Berberine Nanocomplex on Cancer Cells"

_cancers, 2024, doi:10.3390/cancers16183184_

Round 1
Reviewer 1 Report
Comments and Suggestions for Authors
Advantages of the Manuscript:
· The manuscript explores the use of a C60-Berberine nanocomplex in sonodynamic therapy, which is a novel approach with potential for improving cancer treatment efficacy.
· The inclusion of 3D spheroid cultures provides a more physiologically relevant model, enhancing the potential impact of the findings on real-world applications.
· The manuscript provides a detailed description of the materials and methods used, which can aid in reproducibility.
· The study's findings could contribute to the development of targeted cancer therapies with fewer side effects, which is a significant advantage.
Disadvantages of the Manuscript:
· The study's focus on only two cell lines limits the generalizability of the findings. A broader range of cancer types should be tested.
· The lack of detailed statistical analysis weakens the reliability of the results. Without robust statistical validation, the conclusions drawn may not be fully supported.
· The discussion section is somewhat superficial and does not fully explore the implications of the findings or compare them with existing literature.
· The manuscript uses a significant amount of technical jargon without sufficient explanation, which may alienate readers who are not specialists in the field.
These points highlight both the strengths and areas for improvement in the manuscript. Addressing the experimental and grammatical errors, expanding the study scope, and enhancing the discussion could significantly strengthen the manuscript.
Experimental Improvements needed:
· The experimental design does not thoroughly explain how the different variables were controlled, particularly in the comparison between 2D and 3D cell cultures. This could lead to inconsistencies in the results.
· The manuscript mentions control groups treated with sterile water, but there is no mention of additional controls, such as untreated groups or groups treated with US alone without the nanocomplex. This could be an experimental oversight.
· There is no mention of the number of replicates or sample size used in the experiments, which is crucial for statistical analysis and validation of results.
· The manuscript lacks a detailed description of the statistical methods used to analyze the data. This is critical to assess the validity and significance of the findings.
· The potential for artifacts, especially in the preparation of the C60-Berberine complex and its application in cell cultures, is not discussed.
· The study only uses two cell lines (HeLa and LLC), which limits the generalizability of the findings across different cancer types.
Comments on the Quality of English Language
Grammatical Errors:
· There are inconsistencies in the use of tenses, such as switching between past and present tense, which can be confusing for the reader.’
· There are minor typographical errors, such as "sonoluminiscence" instead of "sonoluminescence," and inconsistent punctuation, particularly with commas and hyphens.
· Some terms are used without clear definition or context, which could confuse readers unfamiliar with the subject matter. For instance, the term "sonosesitizer" should be corrected to "sonosensitizer" and clarified.
Reviewer 2 Report
Comments and Suggestions for Authors
In the present work, Radivoievych et al reported the use of C60-Berberine hybrid nano complex for the SDT of cancer. While this is an interesting work that can be published in Cancers, some additional data is also necessary.
1) The material characterization, for example, the morphology of the nano complex and the berberine-to-C60 ratio in the formula.
2) For the use of C60 in PDT, the superoxide anion instead of singlet oxygen is often identified as the ROS. It will be highly useful if the authors are able to identify the actual ROS by spectroscopic means. Otherwise, the conversion of O2 to ROS in Figure 1 may be confusing.
3) Cell staining by DCFH or SOSG might also be helpful.
4) Minor issues: For the definition of Fullerene C60 (C60), I think only C60 is sufficient; for the hybridization state of C60, the average is 2.3, and the number format of ‘2,3’ in line 95 is misleading.
Round 2
Reviewer 2 Report
Comments and Suggestions for Authors
The authors have carefully addressed my concerns and I, therefore, recommend the publication of the manuscript in its current form.